# Pigmented Microbial Extract (PMB) from *Exiguobacterium* Species MB2 Strain (PMB1) and *Bacillus subtilis* Strain MB1 (PMB2) Inhibited Breast Cancer Cells Growth In Vivo and In Vitro

**DOI:** 10.3390/ijms242417412

**Published:** 2023-12-12

**Authors:** Deepa R. Bandi, Ch M. Kumari Chitturi, Jamuna Bai Aswathanarayan, Prashant Kumar M. Veeresh, Venugopal R. Bovilla, Olga A. Sukocheva, Potireddy Suvarnalatha Devi, Suma M. Natraj, SubbaRao V. Madhunapantula

**Affiliations:** 1Department of Applied Microbiology, Sri Padmavathi Mahila Viswavidyalayam, Tirupati 517502, Andhra Pradesh, India; deepareddybandi07@yahoo.com (D.R.B.); drsuvarnaspmvv@gmail.com (P.S.D.); 2Department of Microbiology, JSS Academy of Higher Education & Research (JSS AHER), Mysore 570015, Karnataka, India; jamunabhounsle@jssuni.edu.in; 3Center of Excellence in Molecular Biology and Regenerative Medicine (CEMR) Laboratory, Department of Biochemistry, JSS Medical College, JSS Academy of Higher Education & Research (JSS AHER), Mysore 570015, Karnataka, India; prashanthkumarmv008@gmail.com (P.K.M.V.); venu.1726@gmail.com (V.R.B.); mnsuma@jssuni.edu.in (S.M.N.); 4Department of Gastroenterology and Hepatology, Royal Adelaide Hospital, Adelaide, SA 5000, Australia; olga.sukocheva@sa.gov.au; 5Special Interest Group (SIG) in Cancer Biology and Cancer Stem Cells (CBCSC), JSS Academy of Higher Education & Research (JSS AHER), Mysore 570015, Karnataka, India

**Keywords:** legumain, breast cancer, pigmented microbial extracts, *Exiguobacterium*, *Bacillus subtilis*

## Abstract

Breast cancer (BC) continues to be one of the major causes of cancer deaths in women. Progress has been made in targeting hormone and growth factor receptor-positive BCs with clinical efficacy and success. However, little progress has been made to develop a clinically viable treatment for the triple-negative BC cases (TNBCs). The current study aims to identify potent agents that can target TNBCs. Extracts from microbial sources have been reported to contain pharmacological agents that can selectively inhibit cancer cell growth. We have screened and identified pigmented microbial extracts (PMBs) that can inhibit BC cell proliferation by targeting legumain (LGMN). LGMN is an oncogenic protein expressed not only in malignant cells but also in tumor microenvironment cells, including tumor-associated macrophages. An LGMN inhibition assay was performed, and microbial extracts were evaluated for in vitro anticancer activity in BC cell lines, angiogenesis assay with chick chorioallantoic membrane (CAM), and tumor xenograft models in Swiss albino mice. We have identified that PMB from the *Exiguobacterium* (PMB1), inhibits BC growth more potently than PMB2, from the *Bacillus subtilis* strain. The analysis of PMB1 by GC-MS showed the presence of a variety of fatty acids and fatty-acid derivatives, small molecule phenolics, and aldehydes. PMB1 inhibited the activity of oncogenic legumain in BC cells and induced cell cycle arrest and apoptosis. PMB1 reduced the angiogenesis and inhibited BC cell migration. In mice, intraperitoneal administration of PMB1 retarded the growth of xenografted Ehrlich ascites mammary tumors and mitigated the proliferation of tumor cells in the peritoneal cavity in vivo. In summary, our findings demonstrate the high antitumor potential of PMB1.

## 1. Introduction

Legumain (LGMN), a member of the C13 family of cysteine proteases, was first identified, isolated, and characterized from moth bean (*Vigna aconitifolia* (Jacq.) Marechal) seeds in the early 1990s [1]. In mammals, LGMN was first reported in pigs as a 34 kDa protein and is well conserved among plants, invertebrate parasites, and mammals [2]. The LGMN gene is located on chromosome 14 at 14q32.12 and encodes a proenzyme of 433 amino acids [3]. It shares 83% homology with the murine LGMN [4]. LGMN is also known as asparaginyl endopeptidase (AEP) or asparaginyl carboxypeptidase (ACP) due to its strict specificity to hydrolyze asparaginyl bonds present in various proteins and peptides [2].

The regulatory role of legumain (LGMN) was first reported in cancers in 2003 [5]. LGMN is an oncogenic protein, which can facilitate cancer progression and metastasis. Recent findings have shown that LGMN is one of the tumor-specific biomarkers expressed not only in malignant cells but also in tumor microenvironment (TME) cells, including tumor-associated macrophages (TAMs). LGMN expression has been reported in breast, colorectal, ovarian, prostate, and gastric cancers [6,7,8,9,10]. It is sparsely expressed by normal tissues, making it an attractive anticancer therapeutic target [5]. LGMN overexpression has been associated with the more invasive and metastatic characteristics of tumor cells, indicating its critical role in cancer progression and spread [11]. Patients with elevated LGMN were marked by poor prognosis with reduced survival rates [12,13]. The oncogenic properties of LGMN are associated with the stimulation of cell proliferation, migration, enhanced drug resistance, inhibition of cell differentiation, apoptosis, and cell cycle arrest [14].

Accumulating empirical evidence supported the testing of LGMN as a therapeutic target for novel anticancer pharmacological agents [15]. Accordingly, targeted inhibition of LGMN by silencing RNAs (siRNAs) resulted in decreased cancer cell proliferation [16]. Proton pump inhibitors (PPIs), such as esomeprazole, omeprazole, and lansoprazole, have been reported to inhibit LGMN and retard cell growth [17]. The synthesis of aza-peptidyl inhibitors of LGMN has been reported [18]. Other LGMN inhibitors have been synthesized and characterized, although no single agent showed clinical success [19]. Potential reasons for clinical trial failure include low specificity and sensitivity, a narrow therapeutic window, and the emergence of drug-resistant tumor cells [19]. Therefore, further investigations are warranted to identify more potent and clinically viable pharmacological inhibitors of LGMN.

Prokaryotic organisms, bacteria, and bacteria-derived products demonstrated various antitumor properties [20]. However, there is considerable concern about the application of bacterial cells for cancer treatment, such as the failure of bacteria to destroy cancer cells and/or prevent mutation-promoting effects [21]. Consequently, the use of whole bacteria for the treatment of cancers is not recommended. Prokaryote-related studies have shifted their focus toward the testing of bacterial extracts as anticancer agents. For instance, potent anticancer molecules were isolated from *Streptomycetes* bacteria [22]. Accordingly, the inhibition of tumor growth by bacterial extracts was reported [23].

In the present study, we have generated pigmented microbial (PMB) extracts PMB1 and PMB2 from marine *Exiguobacterium* species MB2 strain and *Bacillus subtilis* strain MB1, respectively. The extracts were assessed to determine their properties as inhibitors of LGMN activity and cancer cell growth. We report that PMB1 selectively inhibits breast cancer (BC) growth, compared to PMB2-induced effects.

## 2. Results

### 2.1. Ethyl Acetate Extract of Exiguobacterium Species MB2 Strain and Bacillus subtilis Strain MB1 Targets LGMN Extracted from Murine Macrophage Cell Line RAW 264.7

Marine pigmented bacteria, which were isolated from samples collected at Maypadu beach (Nellore, Andhra Pradesh, India), were identified as *Exiguobacterium* species MB2 strain and *Bacillus subtilis* strain MB1 by 16S rRNA sequencing with the Gen Bank ID MH910696.1 and MH910670.1, respectively. *Exiguobacterium* species MB2 strain (PMB1) forms orange-yellow pigmented colonies, whereas *Bacillus subtilis* strain MB1 (PMB2) forms cream-colored colonies on Zobell marine agar medium as depicted in the graphical abstract.

The generated microbial extracts yielded pharmacological agent-rich fractions, which induced a dose-dependent reduction in LGMN activity in murine macrophage cell line RAW 264.7 (Figure 1). The significant inhibition of LGMN activity was observed for all tested concentrations (from 31.25 µg/mL to 2000 µg/mL) of the extract. The IC-50 values of PMB1 and PMB2 extracts were 304 µg/mL and 163.9 µg/mL, respectively. Iodoacetamide (IA), a cysteine protease inhibitor, was used as an inhibitor of the enzyme activity (Figure 1). IA showed >96% inhibition at the concentration of 750 µM. In summary, we have found that PMB1 and PMB2 extracts contain molecules that can inhibit LGMN activity. Accordingly, we tested the effects of these extracts using in vitro and in vivo models.

### 2.2. PMB1 and PMB2 Extracts Inhibited BC Cell Growth but Were Inefficient in Normal Lung Epithelial Cell Line BEAS-2B

BC cell lines MDA-MB-468, MDA-MB-231, MCF-7, and a normal cell line BEAS-2B were treated with PMB1 and PMB2 extracts (concentration range: 15.62 µg/mL to 2000 µg/mL) for 24 h and 48 h. The number of viable cells was determined using an SRB assay. Diallyl disulfide (DADS) was used as a positive control. Time- and concentration-dependent decreases in MDA-MB-468 cell viability were observed in cells treated with PMB1 and PMB2 (24 h and 48 h) (Figure 2). A concentration-dependent effect (31.25 µg/mL to 500 µg/mL) of PMB1 extract was detected in MDA-MB-468 cells after 48 h of treatment. A further increase in the concentration to 2000 µg/mL did not show a linear increase in cell death. The highest percentage inhibition (51.6%) of growth was observed after 48 h of treatment with 2000 µg/mL of PMB1 (Figure 2A). PMB2 extract treatment also showed time- and dose-dependent cytotoxicity in MDA-MB-468 cells. A higher cytotoxicity was observed with both extracts after prolonged stimulation (48 h) compared to 24 h of treatment. The highest percentage of inhibition (42.2%) was observed with 2000 µg/mL PMB2 (Figure 2B). IC50s of PMB1 and PMB2 in the MDA-MB-468 cell line at 48 h of treatment were 795 µg/mL and >2000 µg/mL, respectively. Therefore, PMB1 is a more potent inhibitor than PMB2.

PMB1 (250 µg/mL and 2000 µg/mL) treatment also stimulated changes in cell morphology. The treated cells were flat, and intercellular spaces increased. However, PMB2 treatment did not change MDA-MB-468 cell morphology up to 250 µg/mL of concentration. The morphological changes were noticed at 2000 µg/mL of PMB2 (Figure 2C). Inhibition of MDA-MB-468 cell proliferation by DADS (positive control) was observed at 1500 µM of the extract (Appendix A).

The cytotoxicity of PMB1 and PMB2 was assessed in the triple receptor negative (TNBC) cell line MDA-MB-231. Prolonged treatment with PMB1 extract (48 h) showed much higher inhibition of cell proliferation compared to a shorter time of exposure (24 h) (Figure 2D). The highest inhibition of cell proliferation (59.7%) was observed at 48 h treatment with PMB1 extract. PMB2 did not stimulate significant cell death at 24 h. About 21.6% of cell viability inhibition was observed after treatment with 2000 µg/mL of the extract (Figure 2E). IC-50 values for PMB1 and PMB2 extracts in MDA-MB-231 cells (48 h treatment) were 939.3 µg/mL and >2000 µg/mL, respectively. The effects in MDA-MB-231 cells were comparable to those observed in MDA-MB-468 cells. Moreover, both PMB1 and PMB2 also induced changes in MDA-MB-231 cell morphology. Increased intercellular gaps and rounded morphology were observed upon treatment with PMB1 (250 µg/mL and 2000 µg/mL) (Figure 2F). However, treatment with PMB2 did not stimulate any morphological changes at a concentration of up to 250 µg/mL. Morphological changes (increased gaps, flat and rounded morphology) were observed in some MDA-MB-231 cells treated with 2000 µg/mL of PMB2 (Figure 2F). In conclusion, PMB1 efficiently inhibited the growth of TNBC cell line MDA-MB-231, compared to PMB2 extract. DADS (1500 µM) also decreased the viability of MDA-MB-231 cells (Appendix A) and inhibited MCF-7 cell proliferation (Appendix A). To determine the inhibitory effect of the microbial extracts in ER/PR-positive BC cells, we used the MCF-7 cell line. MCF-7 cells were exposed to 2000 µg/mL of PMB1 or PMB2. The extracts reduced the cell viability of BC cells by 49.0% and 47.6%, respectively (Figure 2G,H). IC-50 values for PMB1 and PMB2 extracts were above 2000 µg/mL. Both PMB1 and PMB2 extracts stimulated cell rounding at the highest concentration tested (2000 µg/mL) (Figure 2I).

To test the cell specificity of the observed effects, normal lung epithelial cell line BEAS-2B (control model of normal cell growth) was treated with PMB1 and PMB2 extracts. PMB1 and PMB2 cytotoxicity in BEAS-2B cells was assessed after 48 h of treatment. PMB1 reduced BEAS-2B cell viability by 15% at 1000 µg/mL (Figure 2J). No inhibition in cell proliferation was observed upon treatment of BEAS-2B with PMB2 (Figure 2K). No significant morphological changes were observed upon treatment with PMB1 and PMB2 extracts (Figure 2L). DADS inhibited BEAS-2B cell growth at 1500 µM (Appendix A). Since PMB2 exhibited lower inhibitory potency compared to PMB1, further studies were carried out only with PMB1 extracts (Appendix A).

### 2.3. PMB1 Inhibited LGMN Activity and Expression in MDA-MB-468 Cells

MDA-MB-468 cells demonstrated the highest LGMN expression and activity, compared to other BC cell lines (Figure 3A; ELISA protein activity assay). The changes in LGMN expression and activity were assessed in MDA-MB-468 cells treated with PMB1 extracts. PMB1 (100, 250, or 1000 µg/mL for 48 h) decreased LGMN expression and activity in MDA-MB-468 cells. PMB1 (250 µg/mL and 1000 µg/mL) significantly (*p* < 0.05) decreased the LGMN levels compared to those in untreated MDA-MB-468 cells (Figure 3A). The decrease in the LGMN expression correlated with a concomitant decrease in its activity (Figure 3A). PMB1 (100 µg/mL, 250 µg/mL, and 1000 µg/mL) inhibited LGMN activity on 50.6%, 72.6%, and 72.4%, respectively, compared to untreated cells (Figure 3B).

### 2.4. PMB1 Inhibited the Migration of BC Cells in a Dose- and Time-Dependent Manner

Active cell migration is one of the important characteristics of malignant cells with high metastatic potency. The genetic ablation of LGMN was associated with the reduced activities of matrix metalloproteases (MMPs) and inhibited cell migration and invasion in cervical cancer cell lines [24]. Accordingly, we tested the ability of microbial PMB1 extract to inhibit MDA-MB-468 cell migration using the scratch assay. The cells were treated with various PMB1 concentrations (500, 1000, and 2000 µg/mL) and photographed before the treatment and after 24 and 48 h. Decreases in the scratch area from 21.2 mm (baseline, 0 h) to 11.2 mm (at 24 h) and 7.8 mm (at 48 h) were observed in untreated cell cultures. In the vehicle-treated cells, the scratch area decreased from 21.5 (0 h) to 14.3 (24 h) and 9.9 mm (48 h). Treatment with DADS (750µM) resulted in a minimal gap closure with marginal decreases in the scratch area from 21.3 mm (0 h) to 19.4 mm (24 h) and 19.0 mm (48 h) (Figure 3C,D). PMB1 extract treatment (500 µg/mL) also prevented the gap closure. In the presence of PMB1, the scratch area decreased from 20.2 mm (0 h) to 17.9 mm (24 h) and 17.3 mm (48 h) (Figure 3E,F). In conclusion, microbial PMB1 extract inhibited MDA-MB-468 cell migration in a dose- and time-dependent manner.

### 2.5. PMB1 Inhibits Angiogenesis

LGMN was shown to contribute to angiogenesis via the remodeling of the extracellular matrix (ECM) [25]. Pharmacological inhibition of LGMN blocked angiogenesis in solid tumors [25]. To test whether PMB1 extract can demonstrate anti-angiogenic potential, triggering LGMN inhibition, we used a chorioallantoic membrane (CAM) assay. The model provides a rapid and reliable assessment of angiogenesis-regulating agents [26]. No changes in blood vessel formation were detected in CAM treated with 0.1% DMSO or untreated control CAM (Figure 4A). Treatment of CAM with PMB1 extracts resulted in the inhibition of blood vessel formation (Figure 4B,C). PMB1 (500 µg/mL for 48 h or 1000 µg/mL for 24 h) induced complete loss of blood vessels (indicated by arrows in Figure 4B,C).

### 2.6. Microbial PMB1 Extract Induced Apoptosis in MDA-MB-468 Cells

Numerous apoptosis-inducing pharmacological agents were designed to eliminate cancer cells [27], although the development of drug resistance prevents complete recovery. In this study, we tested the ability of PMB1 extracts to induce apoptosis. Cultured MDA-MB-468 cells were treated with increasing concentrations of microbial extract PMB1 (500, 1000, and 2000 µg/mL) for 48 h and subsequently stained with fluorescent dyes. All control cells (untreated, vehicle control (VC) (1% DMSO)-treated, and positive control (DADS 750 µM)-treated) were stained with the fluorescent dye. The percentage of apoptotic cells in control (untreated), VC, and DADS-treated cells was 3.59, 5.29, and 19.4%, respectively (Figure 5A). Analysis of data indicated nonsignificant (*p* > 0.05) increases in apoptotic cell numbers in VC. A positive control showed a significant (*p* < 0.05) increase in the percentage of apoptotic cells compared to control/VC (Figure 5A). PMB1 (500 µg/mL, 1000 µg/mL, and 2000 µg/mL for 48 h) treatment induced apoptosis in 7.35, 12.9, and 13.6% of MDA-MB-468 cells, respectively (Figure 5B). Higher concentrations of PMB1 extracts (1000 µg/mL and 2000 µg/mL) significantly (*p* < 0.05) increased the percentage of apoptotic cells compared to the control/VC (Figure 5B,C).

### 2.7. PMB1 Extracts Induced Cell Cycle Arrest in MDA-MB 468 Cells

Apoptosis is often accompanied by the induction of cytostasis (cell cycle arrest) [28]. Cytostasis can be measured microscopically by monitoring the changes in cell morphology and counting the number of mono-, bi-, and multinucleated cells [28]. MDA-MB-468 cells were treated with PMB1 extracts (100 µg/mL, 250 µg/mL, and 1000 µg/mL for 48 h), and the cytostasis was assessed using trypan blue staining. All plates were visually evaluated for medium utilization. The media colors showed less consumption of nutrients in the PMB1-treated group, compared to the control/VC (Appendix A).

The cell morphology was also assessed under the inverted microscope. The morphological changes (the predominance of needle-shaped cells) were detected in the cells treated with PMB1 (250 and 1000 µg/mL) (Figure 5D). Cell number was reduced in 1000 µg/mL PMB1-treated wells, compared to 250 µg/mL treatment. Notably, rounded cells were not observed in the presence of 250 µg/mL PMB1. Analysis of cell number showed a nonsignificant (*p* > 0.05) decrease in 100 µg/mL (4.15 × 10^5^ cells) and 250 µg/mL (3.15 × 10^5^ cells)-treated groups, compared to the control (4.99 × 10^5^ cells) and VC (5.45 × 10^5^ cells). The decrease in cell number in the 1000 µg/mL (1.2 × 10^5^ cells)-treated group was found to be significant (*p* < 0.05) (compared to control/VC) (Figure 5E). Assessment of cell viability (MTT assay) did not indicate any significant changes (*p* > 0.05). Cell viabilities in PMB1-treated groups were 96.5% (100 µg/mL), 95.8% (250 µg/mL), and 95.7% (1000 µg/mL), compared to the control (97.7%) and VC (98.0%) (Figure 5F). In conclusion, lower concentration of PMB1 extract (250 µg/mL and 500 µg/mL) stimulated the cell number decreases due to the cytostatic effects of this agent (Figure 5F).

### 2.8. PMB1 Retarded Cancer Growth in Swiss Albino Mice In Vivo

We validated the observed effects using xenograft animal models in vivo [29]. The anticancer efficacy of PMB1 was tested using EAC xenografts in Swiss albino mice (Appendix A). EACs are murine mammary adenocarcinoma cells lacking H-2 histocompatibility genes. EAC cells can proliferate spontaneously in mice and develop solid tumors when injected subcutaneously or as a liquid tumor in the peritoneum [30]. These models are widely used [31].

#### 2.8.1. PMB1 Extract Blocked Xenografted EAC Proliferation (Liquid Cancer Model)

Different groups of female Swiss albino mice were pretreated with PMB1 extracts (50 mg/kg and 100 mg/kg) or cisplatin (3.5 mg/kg, i.p., used as a positive control) for 3 days prior to the introduction of a tumor. Four days later, 1.0 × 10^6^ viable EAC cells were injected. PMB1 treatment was initiated from day 6 and continued for 12 days. Three animals from each group were sacrificed on day 20. Body mass was recorded every alternate day over the treatment period. PMB1 (50 mg/kg) or cisplatin (3.5 mg/kg) induced significant decreases in body mass (Appendix A). The maximum body mass reduction was observed with PMB1 treatment (Figure 6A). PMB1 concentration higher than 50 mg/kg (i.e., 100 mg/kg) was found lethal as two mice (out of six mice) died after the second dose was administered.

#### 2.8.2. PMB1 Reduced the Peritoneal Fluid Volume and Decreased Viability and Cell Number of EAC Cells

The peritoneal fluids were collected immediately after animal sacrifice. Tumor volume, number of tumor cells, and cell viability were assessed. Tumor volumes in control (26.0 mL), cisplatin (16.3 mL), and PMB1 (7 mL)-treated animals were significantly different (*p* < 0.05) (Figure 6B). PMB1 extract (50 mg/kg) induced a significant (73.2%; *p* < 0.05) reduction in the viability of EAC cells collected from peritoneal fluid, compared to control (93%) and cisplatin (80.5%) (Figure 6C). The extract also stimulated a significant (*p* < 0.05) reduction in the tumor cell number (75.1 × 500,000 cells/mL) compared to controls (113.1 × 500,000 cells/mL) or cisplatin (52.6 × 500,000 cells/mL)-treated tumors (Figure 6D). The cancer cells that were collected from the peritoneum were subjected to ethidium bromide and acridine orange staining (apoptosis assay). The extract stimulated a significant dose-dependent increase (*p* < 0.05) in apoptosis. The percentages of apoptotic cells in the control (0.8%), cisplatin (15.7%), and PMB1 (9.6%) were significantly different (*p*< 0.05) (Figure 6E). The staining pattern is shown (Figure 6F). The percentage of apoptotic PMB1-treated cells was lower compared to the cisplatin-treatment group.

#### 2.8.3. PMB1 Inhibited Angiogenesis but Increased Survival of Mice

We tested PMB1-induced effects on angiogenesis in Swiss albino mice bearing EAC cells [31]. The peritoneum from each mouse was collected, and the number and size of blood vessels were registered. PMB1 (50 mg/kg of body mass) inhibited the formation of blood vessels (Appendix A). Following this, we tested whether PMB1 influences the survival of EAC-bearing mice. Using Kaplan-Meier survival analysis, we determined median survival time as follows: control mice = 22 days; cisplatin (3.5 mg/kg) = 28 days; and PMB1 (50 mg/kg) = 30 days. Our data indicates that PMB1 extended the survival period compared to cisplatin-treated and control mice (Figure 6G).

#### 2.8.4. Low PMB1 Toxicity in Liver and Kidneys

The liver and kidneys are the major drug metabolizing organs [32]. To identify the toxicity of PMB1 extract in these organs, the liver and kidney tissues were excised from the control, positive control, and PMB1-treated mice. H&E staining of the paraffin-embedded tissue was used to assess the tissue morphology (Figure 6H). Kidney staining showed normal morphology. No inflammation or necrosis was observed. Glomeruli showed normal histology. All the sacrificed animals from the control showed normal liver architecture and histology. Furthermore, no vacuolar degeneration or blood sinusoids were found during the liver assessment. Hepatocytes did not indicate any abnormalities. In cisplatin-treated mice, the liver showed focal area capsular necrosis with chronic inflammatory cells indicating mild toxic effects. PMB1 (50 mg/kg) extract treatment was marked by subcapsular lymphocytes and occasional focal lymphocyte aggregation with a mild area of necrosis. Our H&E data showed no significant toxicity associated with PMB1 treatment in vivo.

#### 2.8.5. PMB1 Inhibited Growth of EAC Solid Cancer In Vivo

To evaluate the effect of PMB1 extract on EAC solid tumor growth (Appendix A, study design), mice were treated with PMB1 extracts (25 and 50 mg/kg body weight) on every alternative day up to the 25th day (Figure 6I). A significant (*p* < 0.05) decrease in tumor weight (1.8–2.9 fold) was observed in PMB1 (25 and 50 mg/kg) treatment groups (Figure 6I,J). The positive control with cisplatin also yielded a 2.9-fold decrease in tumor mass.

### 2.9. GC-MS Analysis of PMB1 Showed the Presence of Bioactive Compounds

The ethyl acetate PMB1 extract was subjected to GC-MS analysis to identify the extract’s components. The major compounds identified belonged to the class of fatty acids, fatty alcohols, esters, phenolic compounds, hydrocarbons, and alkaloids (Table 1). Prior studies have demonstrated the anticancer properties of phenolic compounds in vitro and in vivo. For instance, phenolic acids are known to exhibit potent anticancer properties by inhibiting key signaling cascades responsible for cancer growth [32,33]. Moreover, fatty acids and their derivatives can alter the cellular oxidative status by enhancing lipid peroxidation, which results in increased cancer cell death [34]. Therefore, it is speculated that the phenolic compounds and fatty acid derivatives found in PMB1 extract are the potential key compounds responsible for anticancer activity.

## 3. Discussion

Drug resistance develops to the majority of currently available anticancer agents [35]. TME contributes to the process of cancer immunoediting and facilitates the development of drug resistance [36]. TME was shown to secrete different cytokines and interleukins responsible for cancer cell survival during radio- and chemotherapy treatments [37]. Therefore, the search for a drug that will target both cancer cells and their TME continues [38]. LGMN is a protein overexpressed both in TME and tumor cells [1] (Figure 1). Accordingly, LGMN was named as a potential anticancer target [39,40]. Seminal findings from prior reports demonstrated the high anticancer potential of LGMN-targeting in BC in vitro and in vivo [40]. We detected a significant increase in the activity of LGMN in BC cells (Figure 3). Therefore, we attempted to extract natural anti-LGMN inhibitors from marine sources and assess their efficiency in BC cells and in murine cancer models.

*Exiguobacterium* species MB2 and *Bacillus subtilis* MB1 strains that were used in this study have not been tested for antitumor activities previously. The bacterial ethyl acetate extracts, PMB1 and PMB2, were evaluated for inhibition of oncogenic cysteine protease LGMN ex vivo. Both extracts significantly inhibited LGMN activity (Figure 1). MDA-MB-468 cells demonstrated the highest LGMN expression and were chosen for further investigations. PMB1 reduced LGMN expression and activity in MDA-MB-468 cells. Supporting our findings, six structurally different carotenoids isolated from the methanolic extract of *Exiguobacterium acetylicum* S01, namely lycopene (Car-I), diapolycopenedioic-acid-diglucosyl-ester (Car-II), β-carotene (Car-III), zeaxanthin (Car-IV), astaxanthin (Car-V), and keto-myxocoxanthin glucoside-ester (Car-VI) demonstrated anticancer activity in HT-29 colorectal cancer cells [41]. The carotenoids inhibited HT-29 cell growth in a dose-dependent manner while having no cytotoxic effect in normal blood monocytes (PBMCs) [41]. Previously, Solberg et al., 2015 reported inhibition in LGMN activity in M2-polarized human monocytes with microbial statins [42]. However, our study is the first to report the anticancer activity of *Exiguobacterium* species (PMB1 extract) in BC cell lines.

Furthermore, PMB1 extracts inhibited BC cell migration in a dose- and time-dependent manner. It has been reported that the knockdown of LGMN in cervical cancer cell lines resulted in the reduction of cell migration/invasion mediated by MMPs [24]. Previously, the growth-regulatory role of LGMN was confirmed in MDA-MB-231 and MDA-MB-435 cell lines using small molecule inhibitors [40]. LGMN was also shown to control angiogenesis via ECM remodeling [25]. Inhibition of LGMN decreased angiogenesis [25]. In the current study, the treatment of CAM with PMB1 resulted in a significant decrease in the number of blood vessels. In support of our findings, Elaiophylin (a C2 symmetry glycosylated 16-membered macrolide from *Streptomyces melanosporus*) potently inhibited the angiogenesis of CAMs without toxic effects in the pre-existing vessels [43]. Another microbial metabolite with two epoxide groups, rhizoxin, exhibited antitubulin activity and inhibited angiogenesis in CAMs [44].

Treatment of MDA-MB-468 cells with PMB1 extract increased the number of cells undergoing apoptosis in a dose-dependent manner. Our results are in agreement with the previous reports. For instance, prodigiosin (red-colored pigment from *Serratia marcescens*) induced apoptosis in DLD-1 and SW-620 human colon adenocarcinoma cells [45]. Violacein (a pigment from *Chromobacterium violaceum*) also enhanced apoptosis in colorectal cancer and BC cells [46,47]. In our study, PMB1 extracts demonstrated anticancer potential, and showed selective toxicity towards BC cell lines, while exhibiting lower toxicity towards the normal lung epithelial cell line. Moreover, treatment of MDA-MB-468 cells with PMB1 resulted in the cytostatic effect, reducing cell number without affecting cell viability. The effect may be mediated by PMB1 extract components that include phenolic acid. Phytochemical analysis of the extract by GC-MS revealed the presence of not only phenolic acids but also esters of fatty acids. Phenolic acids are known to exhibit potent anticancer properties and inhibit key growth-regulating effectors, including members of the PI3K-Akt pathway, HDAC, and others [32,33]. However, further studies are warranted to determine the structure and activity of predominant phenolic compounds present in the extract.

The detected anticancer efficacy of PMB1 extracts was confirmed in vivo using murine cancer xenograft models. Treatment with PMB1 microbial extract inhibited the growth of EAC solid tumors in vivo. Administration of PMB1 extracts also resulted in the reduction of EAC cell numbers, indicating the retarded liquid tumor growth in mice. The treatment also increased the number of EAC cells undergoing apoptosis, while angiogenesis was inhibited. Liquid tumors (like the EAC model) have circulatory tumor cells, which are similar to the metastatic cells in the circulation in humans [48]. This model was also used as a preliminary metastatic model for the assessment of pharmacological agents. Therefore, we speculate that PMB1 extracts may demonstrate antimetastatic capacity, although this statement requires future investigations.

Finally, we found that PMB1 extract is nontoxic to the animal as no significant changes were observed in the vital organ (liver and kidneys) morphology and architecture. While reducing tumor burden, the extracts improved the mean survival of mice. Our data indicates that PMB1 extract may provide safe antitumor effects for a longer duration. This study is the first to report anticancer activity of *Exiguobacterium* species (PMB1) in animal models. However, in a similar study by Islam et al., 2014, the petroleum ether extract/bacterial metabolite from *Corynebacterium xerosis* significantly decreased cancer cell growth and tumor weight and increased the life span of EAC-bearing mice [49].

## 4. Materials and Methods

### 4.1. Sample Collection and Isolation of Bacteria

PMB1 and PMB2 were isolated from the bacterial samples collected from Maypadu Beach, Nellore, Andhra Pradesh, India. The bacteria were isolated, cultured, and characterized at the Department of Applied Microbiology, Sri Padmavati Mahila Visvavidyalayam, Tirupati, Andhra Pradesh, India. Bacteria were identified as *Exiguobacterium* species MB2 strain (Gen Bank ID: MH910696.1) and *Bacillus subtilis* strain MB1 (Gen Bank ID: MH910670.1), respectively, using 16S rRNA sequencing as described previously [50]. The sequence details are submitted to NCBI Gen Bank (*Exiguobacterium* sp. strain MB2 16S ribosomal RNA gene, partial sequence—Nucleotide—NCBI (nih.gov) and *Bacillus subtilis* strain MB1 16S ribosomal RNA gene, partial sequence—Nucleotide—NCBI (nih.gov)).

### 4.2. Preparation of Marine Microbial Extract

*Exiguobacterium* species MB2 and *Bacillus subtilis* MB1 were used for the agent extraction. The bacterial strains were inoculated into the sterile 1.0-liter nutrient broth (Peptone 10 g/L, beef extract 10 g/L, and NaCl 5 g/L, pH 7.3) under aseptic conditions. The inoculated culture was incubated at 28 °C for 7 days at 100 RPM in a shaker incubator. After 7 days of incubation, the broth was centrifuged at 10,000 RPM for 30 min at 4 °C and the supernatant was collected. The collected supernatant was mixed with an equal volume of ethyl acetate, and the organic layer was separated. The separated organic layer was dried under a vacuum, and stock solutions (200 mg/mL) were prepared by dissolving in cell culture grade DMSO. Further dilutions were prepared either in PBS or in a cell culture medium [51].

### 4.3. LGMN Inhibition Assay

LGMN inhibition by microbial extracts PMB1 and PMB2 was evaluated as detailed in Berven et al. 2013 [52]. RAW 264.7 protein cell lysate was prepared using RIPA buffer, consisting of 50 mM Tris HCl, 150 mM NaCl, 1.0% (*v*/*v*) NP-40, 0.5% (*w*/*v*) sodium deoxycholate, 1.0 mM EDTA, 0.1% (*w*/*v*) SDS, and 0.01% (*w*/*v*) sodium azide at a pH of 7.4. The protein lysate was used as a source of LGMN [52]. RAW 264.7 protein lysate (20 µL from a stock of 1.25 mg/mL) was pre-incubated at 37 °C with increasing concentrations of PMB1 and PMB2 extracts (31.25 µg/mL to 2000 µg/mL) for 1.0 h. Untreated RAW 264.7 lysate was used as a control for total LGMN activity. The final volume was made up to 120 µL with an LGMN assay buffer (39.5 mM citric acid pH 5.8, 121 mM Na_2_HPO_4_, 1 mM Na_2_EDTA, 0.01% CHAPS, and 1 mM DTT). After 1.0 h of incubation, 40 µL LGMN substrate (40 µM Z-Ala-Ala-Asn-AMC) was added. The liberated fluorescence was measured immediately (considered as 0 min) and for 90 min in a multimode plate reader (Perkin Elmer, Germany) with 360 nm excitation and 460 nm emission [53]. The LGMN activity was calculated as µmoles of 7-amino-4-methylcoumarin (AMC) produced per minute per mg of protein. The % inhibition in LGMN activity was calculated as follows: % inhibition = 100 × (1 − LGMN activity of sample/LGMN activity of control).

### 4.4. Evaluation of the Efficacy of Microbial Extracts for Inhibiting the Proliferation of BC Cells In Vitro

BC cell lines MDA-MB-468 and MDA-MB-231 (oestrogen receptor negative (ER^−^), progesterone receptor negative (PR^−^), human epidermal growth factor receptor negative (HER2^−^); and MCF-7 (ER^+^, PR^+^, HER2^−^) were procured from National Center for Cell Science (NCCS), Pune, Maharashtra, India and cultured in Dulbecco’s Modified Eagle Medium (DMEM) supplemented with 10% FBS, 1% glutamine, and 1% penicillin-streptomycin in a CO_2_ incubator (maintained at 5% CO_2_) (Thermo Fisher Scientific, Waltham, MA, USA), maintained at 37 °C with 95% relative humidity. An immortalized, nontumorigenic, lung epithelial cell line of human origin, i.e., BEAS-2B was provided by Dr Rajeshkumar Thimmulappa, Professor of Biochemistry, JSS Medical College, JSS Academy of Higher Education & Research, Mysore, Karnataka, India, and was cultured in alpha-MEM supplemented with 10% FBS and 1% penicillin-streptomycin. The confluent cultures were subcultured by trypsinization followed by seeding the cells into culture flasks (T-25 or T-75).

#### 4.4.1. Determination of Antic Activity of Extract by SRB Assay

The anticancer activity of microbial extracts was measured according to Madhunapantula et al., 2008 [54]. In brief, 1 × 10^4^ BC cells (MDA-MB-468, MDA-MB-231, and MCF-7) and a normal cell line (BEAS-2B) in 100 µL DMEM, supplemented with 10% FBS were seeded in a 96-well plate and incubated at 37 °C in a cell culture incubator (maintained at 5% CO_2_ with 90% relative humidity). After ~36 h, the cells were exposed to increasing concentrations of microbial extract PMB1 and PMB2 (15.62 µg/mL to 2000 µg/mL) for 24 h and 48 h. Inhibition of cell proliferation was measured using sulforhodamine-B assay (SRB assay) as detailed in Bovilla et al., 2021 [55]. In brief, the SRB assay was carried out by first fixing the cells with cold 50% TCA (Trichloroacetic acid) at 4 °C for 1 h, followed by washing the fixed cells gently under running tap water and air-drying. Staining of fixed cells was performed by adding SRB solution (0.04%) and incubating at room temperature for 1 h. The unbound excess dye was removed by gently washing the plates with 1% acetic acid. The plates were air-dried again, and Tris base solution (10 mM) was added to solubilize protein-bound dye. The absorbance was measured at 510 nm and % cell viability was calculated as follows: % Viability = (OD of test − OD of media blank)/(OD of vehicle-treated cells − OD of media blank) × 100.

#### 4.4.2. Evaluation of LGMN Expression and Activity in the Extract-Treated MDA-MB-468 Cells

MDA-MB-468 cells (2 × 10^6^) were plated in 100 mm Petri plates (P-100) to assess the LGMN expression [52,53]. When the cells reached about 50–60% confluency (~26 h–28 h later), the culture plates were washed with PBS and 8.0 mL of the fresh cell culture media was added. The plates were subsequently treated with increasing concentrations of microbial extract PMB1 (100 µg/mL, 250 µg/mL, and 1000 µg/mL) for 24 h and 48 h to assess the time- and dose-dependent effects on the LGMN expression and activity. Untreated cells were used as controls. Cell lysates were collected by the addition of RIPA buffer, and total protein concentration was estimated using the BCA method as detailed in the supplier manual (Thermo Fisher Scientific, Waltham, MA, USA). The LGMN activity was assessed as detailed before, and the LGMN protein expression was quantified by ELISA (RayBio^®^, Norcross, GA, USA).

#### 4.4.3. PMB1 Effects on MDA-MB-468 Cell Migration Using Scratch Assay

The effect of PMB1 extract on MDA-MB-468 cell migration was evaluated using the scratch assay [56]. MDA-MB-468 cells (4 × 10^3^) were plated in a 12-well plate and, after 70% confluency (~30 h), a scratch was introduced at the center of the well using a plastic 10 µL pipette tip. The resulting gap was equal to the outer diameter of the end of the tip (0.3 mm). The media was removed, and wells were washed with PBS to remove the floating cells. The wells were replenished with fresh medium containing 10% FBS. The wells were further treated with DMSO (vehicle control (VC), 1%), and microbial extract PMB1 (500 µg/mL, 1000 µg/mL, and 2000 µg/mL) and diallyl disulfide (DADS, 750 µM dissolved in 1% DMSO). DADS (750µM) was used as a positive control for the inhibition of cell migration [57]. The cell-free area was calculated in plates with the treated cells and compared with the scratch area in untreated controls. The cells were observed under an inverted microscope at 0 h, 24 h, and 48 h and the images were captured for quantifying the gap area using ImageJ software, https://imagej.net/ (accessed on 19 October 2023).

#### 4.4.4. Detection of Apoptosis by Acridine Orange and Ethidium Bromide Staining

Apoptosis was assessed using acridine orange and ethidium bromide staining methods [58]. MDA-MB-468 cells (0.5 × 10^6^) were plated in 6-well plates and, after ~30 h, exposed to increasing concentrations of microbial extract PMB1 (500 µg/mL, 1000 µg/mL, and 1500 µg/mL) and DADS (750 µM) for about 48 h. The control and treated cells were trypsinized and mixed thoroughly to obtain a single-cell suspension. Trypsin was neutralized by the addition of a complete medium, and 20 µL cell suspension was incubated with 10 µL ethidium bromide (100 µg/mL) and 10 µL of acridine orange (100 µg/mL) mixture for 10 min. The live cells appeared green in color. The apoptotic cells appeared red when the stained cells were observed under a fluorescent microscope. The cells were imaged using the fluorescence microscope (Olympus, Shinjuku-ku, Tokyo 163-0914, Japan) using tetramethylrhodamine isothiocyanate (TRITC) and fluorescein isothiocyanate (FITC) filters. The average green and orange cell numbers were used for the calculation of % apoptosis as follows: % apoptosis = (Number of orange cells/Total number of cells) × 100.

#### 4.4.5. Assessment of PMB1 Effects on the Viability of MDA-MB-468 Cells

MDA-MB-468 cells (0.5 × 10^6^) were plated in 6-well plates and, after ~30 h, exposed to increasing concentrations of PMB1 (100 µg/mL, 250 µg/mL, and 1000 µg/mL) and 1% DMSO (vehicle control) for about 48 h. The cells were trypsinized and subjected to a trypan blue exclusion test for assessment of cell viability [59]. The dead and live cells were counted using a Neubauer chamber, and the cell viability % was calculated.

### 4.5. Angiogenesis Assay with Chick Chorioallantoic Membrane (CAM)

The anti-angiogenic potential of PMB1 was evaluated using chick chorioallantoic membrane (CAM) assay as described previously [60]. No ethical approval was obtained to conduct the CAM assay as the chick embryo is not considered a living animal till day 17 of development [61]. Fertilized eggs (day 1) were procured from Malavalli Poultry, Mandya, Karnataka, India. The surface of the eggs was wiped with 70% ethanol and incubated at 37 °C incubator for the development of blood vessels. On day 7, the surface of the egg was wiped with 70% ethanol. Using a single-edged razor blade, a square (6.25 cm^2^) was created, and the egg was punctured gently against the eggshell until the shell came out to produce an opening. Eggs with viable embryos were selected, and 20 µL of microbial extracts PMB1 (250 µg/mL, 500 µg/mL, and 1000 µg/mL) was added slowly in the opening of eggs using a micropipette. DADS (750 µM) was used as the positive control, and 0.1% PBS was used as the vehicle control. The gap was closed using tape, and the eggs were placed back in the incubator. The effect on angiogenesis was observed after 24 and 48 h by measuring the blood vessel density/size.

### 4.6. Tumor Xenograft Models in Swiss Albino Mice

PMB1 effects were evaluated in vivo using liquid and solid tumor models in Swiss albino mice using Ehrlich ascites carcinoma (EAC) [61].

#### 4.6.1. PMB1 Extract Efficacy in EAC Liquid Tumor Model In Vivo

EAC liquid tumor study was approved (P 11 -281/2018) by the Institutional Animal Ethics Committee, JSS College of Pharmacy, JSS Academy of Higher Education & Research, Mysore, Karnataka, India. A total of 50 female Swiss albino mice of 6–8 weeks old weighing around 25–28 g were selected and divided into nine groups (Appendix A). All the groups were given two doses of PMB1 extract intraperitoneally (i.p.) prior to the inoculation of tumor cells (day 1 and day 3) as described in the study design (Appendix A). Tumor cells collected from EAC-bearing mice were diluted with PBS, and viable cells were counted using the trypan blue exclusion method [62]. The experimental mice were later injected intraperitoneally with 1 × 10^6^ viable cells to develop liquid tumors (day 4) as shown in (Appendix A). The treatment with microbial extract PMB1 (50 and 100 mg/kg body weight) was initiated on day 6 and continued till day 18 by administering the extracts on alternative days. Body weight was recorded every other day. On the 20th day, three mice from the microbial extract PMB1-treated group were sacrificed by CO_2_ asphyxiation followed by cervical dislocation. The remaining animals from all the groups were maintained without the treatment for the assessment of mean survival time. The ascites collected were used for the estimation of cell viability using the trypan blue exclusion method. Apoptosis was assessed using acridine orange and ethidium bromide staining. The blood vessels were photographed to study the impact on angiogenesis. The liver and kidney from the sacrificed animals were collected for tissue histopathology with hematoxylin and eosin (H&E) staining [63].

#### 4.6.2. PMB1 Extract Efficacy in EAC Solid Tumor Model In Vivo

PMB1 extract efficacy was tested using EAC solid tumors in Swiss albino mice, as detailed previously [64]. Swiss albino mice (age: 6–8 weeks) weighing around 26–30 g were divided into nine groups (Appendix A). Five million (5 × 10^6^) viable tumor cells collected from EAC-bearing mice were injected into the right thigh tissue of 36 experimental animals (study design: Appendix A). Four animals were not injected with EAC cells and were maintained as normal controls. The tumor volume was measured using vernier calipers once every five days. On the 12th day, the mice were injected with PMB1 (25 mg/kg and 50 mg/kg body weight). The injection was repeated every alternate day intraperitoneally. On the 25th day, four mice from each group were sacrificed. Tumors were excised, and their weight determined. The tumor sizes were recorded and compared between groups.

### 4.7. Analysis of PMB1 Extract by Gas Chromatography-Mass Spectrometry (GC-MS)

The ethyl acetate extract of PMB1 was analyzed using GC-MS (GC-7890A/MS-5975 C model, Agilent Technologies, Santa Clara, CA, USA) equipped with HP-5 MS column (30 m × 0.25 mm × 0.25 µm). Helium gas was used as a carrier with a flow rate of 1 mL/min. The initial temperature was set at 50 °C and raised to 200 °C at the rate of 10 °C/min. The mass spectrophotometric detector was operated in electron impact ionization mode with an ionizing energy of 70 eV and scanning mode set from *m*/*z* 40–500 Da. The ion source temperature was 200 °C. GC-MS detection of various constituents of PMB1 was based on computer evaluation of mass spectra of samples through the National Institute Standard and Technology (NIST), comparison of peaks, retention time, and computer matching of the characteristics and fragmentation patterns of mass spectra for particular compounds.

### 4.8. Statistical Analysis

All experiments were conducted with at least three replicates (intra-experimental). The results were expressed as mean (of three independent experiments) ± SEM, calculated using GraphPad prism version 6.0. Data was compared using the Student “*t*” Test or Dunnetts multiple comparisons (DMC) test (One-way ANOVA). The “*p*” value of <0.05 was considered significant.

## 5. Conclusions

The sequential evaluation of microbial extracts ex vivo, in vitro, and in vivo showed the high potential of PMB1 extract to decrease BC cell proliferation and migration. The effects are potentially mediated by inhibition of LGMN activity and expression. Furthermore, the extract induced cancer cell apoptosis and blocked angiogenesis in EAC xenograft models. Future experimental studies are warranted to identify and characterize the exact LGMN-inhibitory compounds from *Exiguobacterium*. Additional studies should aim to evaluate PMB1-derived compound(s) pharmacokinetics, dynamics, and the mechanisms of LGMN inhibition.

## Figures and Tables

**Figure 1 ijms-24-17412-f001:**
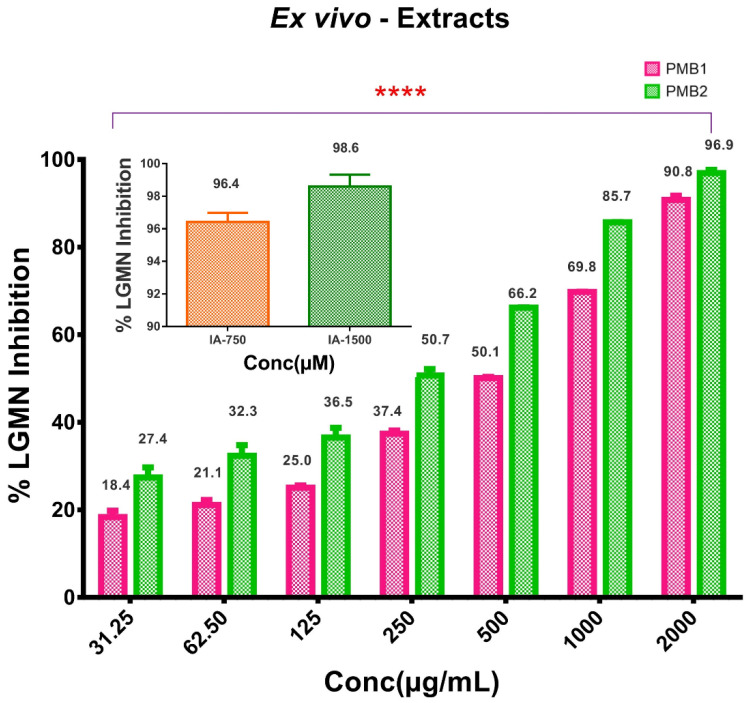
Microbial extracts PMB1 and PMB2 inhibited LGMN activity ex vivo. Inhibition of LGMN activity by microbial extracts PMB1 and PMB2 was performed as detailed in the Methods (Section 4). A dose-dependent increase in the percentage inhibition of LGMN activity was observed with PMB1 and PMB2 extracts. IA (positive control) inhibited LGMN activity by >90% at 750 µM and 1500 µM concentrations (Insert). Statistical analysis was performed using the DMC Test. ****—indicates *p*-value < 0.001.

**Figure 2 ijms-24-17412-f002:**
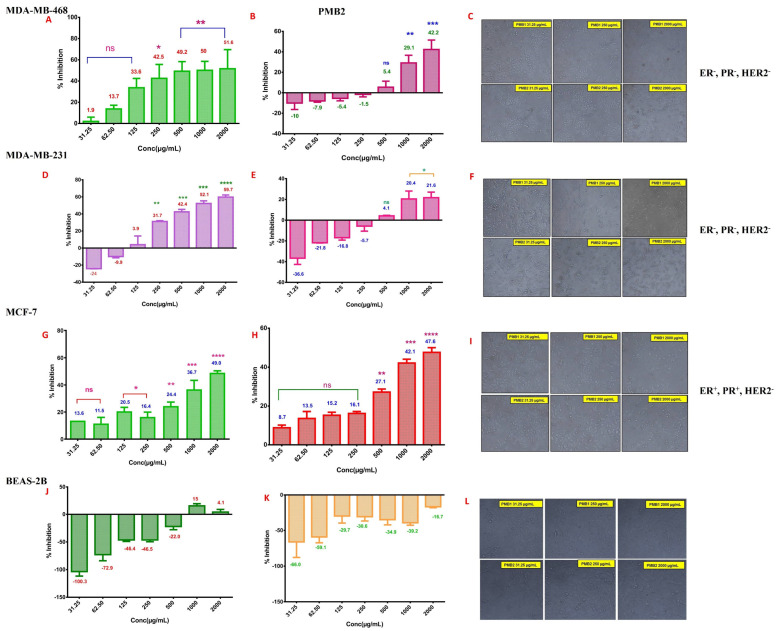
Microbial extracts PMB1 and PMB2 inhibited BC cell growth without affecting the viability of normal lung epithelial cells BEAS-2B. PMB1 more effectively reduced the viability of MDA-MB-468 (**A**), MDA-MB-231 (**D**), and MCF-7 (**G**) BC cells compared to PMB2 effects (**B**,**E**,**H**). Photomicrograph analysis at 10× showed dose-dependent changes in the morphology of BC cells treated with PMB1 (**C**,**F**,**I**) & treatment of normal human lung epithelial cell line BEAS-2B with PMB1 increased cell number more evidently compared to PMB2 (**J**–**L**). Dimethyl sulfoxide (DMSO; 1%) was used as the vehicle control. Statistical analysis was performed using the DMC test. *, **, ***, **** indicates *p* < 0.05, <0.01, 0.001, 0.0001, respectively; “ns” indicates “non significant”.

**Figure 3 ijms-24-17412-f003:**
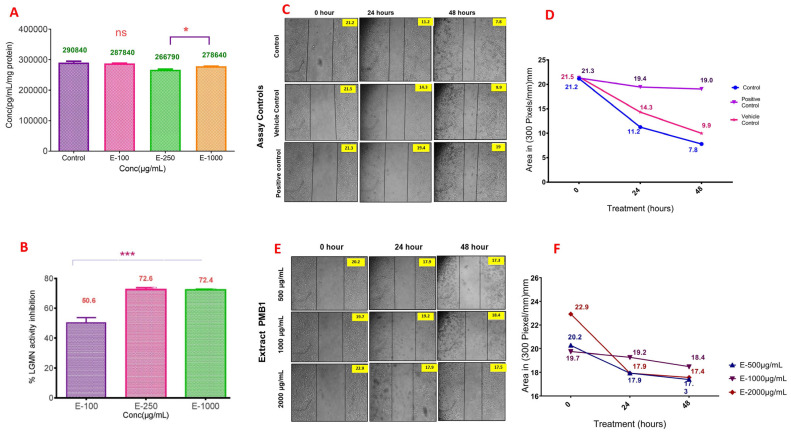
Microbial extract PMB1 inhibited LGMN activity and reduced the migration of TNBC cell line MDA-MB-468. (**A**) Expression of LGMN in MDA-MB-468 cells treated with microbial extract PMB1 for 48 h showed a significant reduction at 250 µg/mL and 1000 µg/mL concentration. (**B**) The inhibition in LGMN activity compared to untreated cells was 50.6, 72.6 and 72.4% in 100 µg/mL, 250 µg/mL, and 1000 µg/mL treated cells, respectively; microbial extract PMB1 mitigated the migration of MDA-MB-468 cells (**C**–**F**). Photomicrographs (30×) of untreated MDA-MB-468 cells and the cells exposed to DMSO (1%), positive control DADS (750 µM), and increasing concentration (500 µg/mL, 1000 µg/mL, and 2000 µg/mL) of PMB1 at 24 and 48 h. PMB1 inhibited the migration of MDA-MB-468 cells, as evidenced by a minimal decrease in the scratch compared to untreated control and vehicle-treated cells. *, ***, indicates *p* < 0.05, <0.001, respectively; “ns” indicates “non significant”.

**Figure 4 ijms-24-17412-f004:**
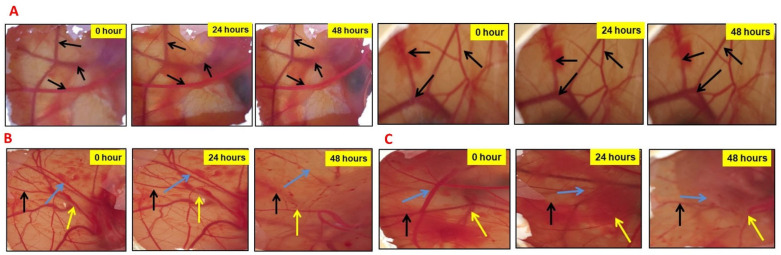
Microbial extract PMB1 inhibited angiogenesis in a dose-dependent manner in chick chorioallantoic membrane (CAM) assay (**A**–**C**) Photograph of CAM exposed to control and vehicle PBS at 0, 24 and 48 h (**A**), 500 µg/mL of microbial extract PMB1 (**B**) and 1000 µg/mL microbial extract PMB1 (**C**). A dose-dependent decrease in the number as well as size of the blood vessels was observed with PMB1 treatment (**C**).

**Figure 5 ijms-24-17412-f005:**
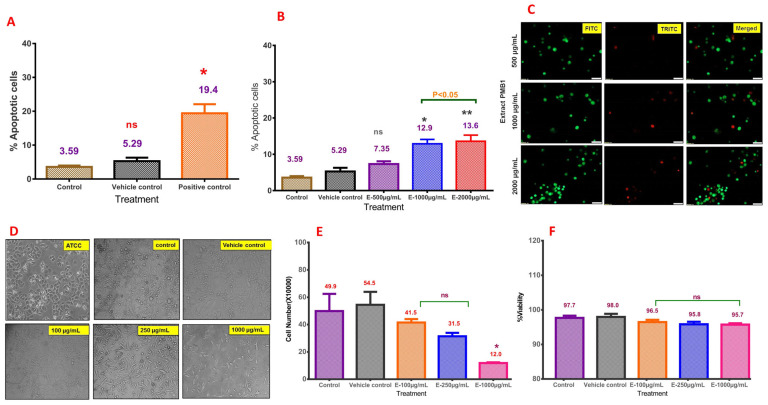
PMB1 induced apoptosis in the MDA-MB-468 cell line. Exposure of MDA-MB-468 cells to vehicle control (VC; 1% DMSO) did not induce any significant apoptosis (**A**); however, treatment with positive control 750 µM DADS moderately elevated cells undergoing apoptotic death. Microbial extract PMB1-treated cells (500 µg/mL, 1000 µg/mL, and 2000 µg/mL) showed a dose-dependent increase in the percentage of apoptotic cells (**B**). Photomicrographs (10×) of cells treated with microbial extract PMB1 (500 µg/mL, 1000 µg/mL and 2000 µg/mL). A dose-dependent increase in the apoptotic cells was observed; Apoptosis detection in microbial extract PMB1-treated (500 µg/mL, 1000 µg/mL, and 2000 µg/mL) MDA-MB-468 cells using ethidium bromide and acridine orange staining. The images obtained using 2 different channels were merged to obtain a combined image, which emitted green and orange cells. The live cells take up acridine orange which stains the cells green, while the apoptotic cells, whose membrane integrity is lost and nucleus is exposed take up the ethidium bromide and appear orange when photomicrographed under fluorescence microscope. (**C**). Morphological changes in the MDA-MB-468 cells upon treatment with microbial extract PMB1 (100 µg/mL, 250 µg/mL, and 1000 µg/mL) for 48 h showed a major difference in the cell size and morphology between control and PMB1-treated samples (100 µg/mL, 250 µg/mL, and 1000 µg/mL) (**D**). Comparison of change in cell number of MDA-MB-468 cells treated with microbial extract (100 µg/mL, 250 µg/mL, and 1000 µg/mL) for 48 h and control using the DMC test (**E**). Comparison of the viability of MDA-MB-468 cells treated with microbial extract (100 µg/mL, 250 µg/mL, and 1000 µg/mL) for 48 h with controls using the DMC test (**F**). *, **, indicates *p* < 0.05, <0.01, respectively; “ns” indicates “non significant”.

**Figure 6 ijms-24-17412-f006:**
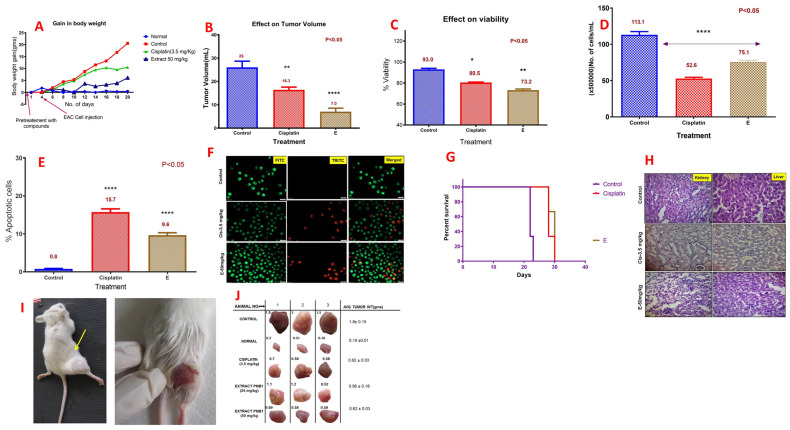
Microbial extract PMB1 exhibited anticancer activity in EAC xenografts in mice in vivo. (**A**) Body weight changes. (**B**) Comparison of the tumor volume on the day of sacrifice (Day 20) between different treatment groups and controls. (**C**) Viability of EAC cells collected from peritoneal fluid. (**D**) A comparison of the number of EAC cells collected from peritoneal fluid of different treatment groups showed a moderate increase in the number of apoptotic cells. (**E**) Comparison of % of apoptotic cells in different treatment groups in the peritoneal fluid of Swiss albino mice treated with PMB1 compared to control untreated EAC mice. (**F**) Apoptosis detection in EAC cells collected from mice peritoneum on the day of sacrifice in control, positive control, and microbial extract PMB1-treated group. Green color indicates live cells whereas the Red color indicates the apoptotic cells (**G**) The Kaplan-Meier survival analysis for the calculation of median survival time showed beneficial outcomes with PMB1-treated mice compared to control animals. (**H**) H&E staining of liver and kidney tissues (20×) in control, cisplatin, and microbial extract PMB1-treated groups. The tumor volume was measured using vernier calipers on every alternative day. (**I**,**J**) Tumors excised from the mice on the 25th day of the treatment. *, **, **** indicates *p* < 0.05, <0.01, 0.0001, respectively.

**Table 1 ijms-24-17412-t001:** Compounds identified in the ethyl acetate extract of PMB1 by GC-MS analysis.

S. No	Retention Time (RT, in Minutes)	Area %	Compound Name and Molecular Weight (Daltons)	Chemical Structure	CAS#
1	8.93	0.65	1-Hexene, 3,5-dimethyl-Molecular weight: 112.2126Other names: 3,5-Dimethyl-1-hexene; 3,5-Dimethylhex-1-ene	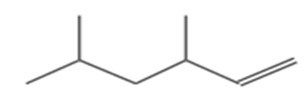	7423-69-0
5-Methyl-1-heptanolMolecular weight: 130.2279Other names: 5-methylheptan-1-ol	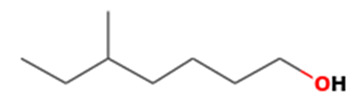	7212-53-5
Cyclopropane, 1-hexyl-2-propyl-, cis-Molecular weight: 168.32Other names: 1-Hexyl-2-propylcyclopropane	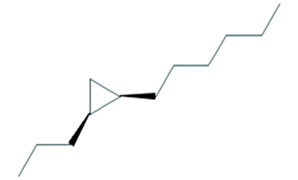	74630-58-3
3-Dodecene, (Z)-Molecular weight: 168.3190Other names: cis-3-Dodecene; (3Z)-3-Dodecene; (Z)-3-dodecene	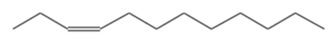	7239-23-8
Cyclopropane, 1-ethyl-2-heptyl-Molecular weight: 168.32 Other names: 1-Ethyl-2-heptylcyclopropane;	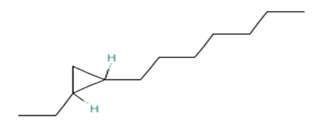	74663-86-8
2	10.09	6.849	Benzaldehyde, 4-methoxy-Molecular weight: 136.1479Other names: p-Anisaldehyde; p-Anisic aldehyde; p-Formylanisole; p-Methoxybenzaldehyde; Anisaldehyde; Aubepine; Crategine; Obepin; 4-Anisaldehyde; 4-Methoxybenzaldehyde; Anisic aldehyde; Anisaldehyde (para); para-Anisaldehyde; NSC 5590; Anisal; Methoxybenzaldehyde	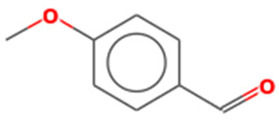	123-11-5
Benzaldehyde, 3-methoxy-Molecular weight: 136.1479Other names: m-Anisaldehyde; m-Methoxybenzaldehyde; 3-Anisaldehyde; 3-Methoxybenzaldehyde; Metamethoxybenzaldehyde	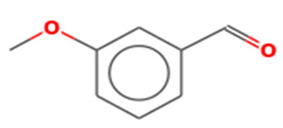	591-31-1
2-(4-Methoxyphenyl)-2-ketoethylamine, PFPMolecular weight: 311.20Other names: 2,2,3,3,3-Pentafluoro-N-[2-(4-methoxyphenyl)-2-oxoethyl]propenamide;N-[2-(4-Methoxyphenyl)-2-oxoethyl]-2,2,3,3,3-pentafluoropropanamide	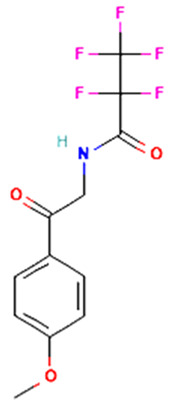	N/A
S-(p-Methoxybenzoyl)thiohydroxylamineMolecular weight: 183.23Other names: S-amino 4-methoxybenzenecarbothioate	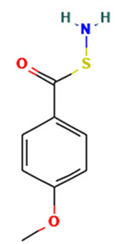	35124-66-4
Benzoic acid, 4-methoxy-, 4-ethylphenyl esterMolecular weight: 256.30Other names: 4-Ethylphenyl 4-methoxybenzoate	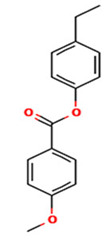	7465-91-0
3	11.69	0.996	3-Tetradecene, (Z)-Molecular weight: 196.3721Other names: (3Z)-3-Tetradecene; cis-3-Tetradecene; (Z)-3-Tetradecene	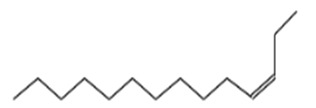	41446-67-7
Acetic acid, trifluoro-, 3,7-dimethyloctyl esterMolecular weight: 254.29Other names: 3,7-dimethyloctyl 2,2,2-trifluoroacetate	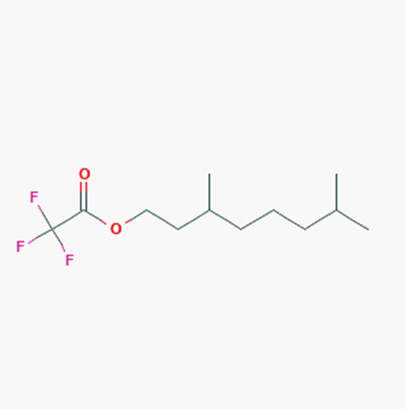	28745-07-5
1-DodeceneMolecular weight: 168.3190Other names: α-Dodecene; n-Dodec-1-ene; Adacene 12; Dodec-1-ene; α-Dodecylene; Dodecylene α-; Dodecene-1; Neodene 12; NSC 12016	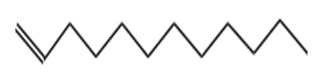	112-41-4
Cyclopropane, 1-ethyl-2-heptyl-Molecular weight: 168.32Other names: 1-Ethyl-2-heptylcyclopropane; Cyclopropane, 1-ethyl-2-heptyl-1-Ethyl-2-heptylcyclopropane	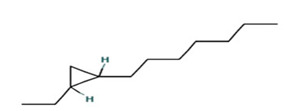	74663-86-8
7-Tetradecene, (E)-Molecular weight: 196.3721Other names: (7E)-7-Tetradecene; (E)-7-Tetradecene; E-Tetradec-7-ene; trans-7-tetradecene	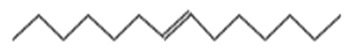	41446-63-3
4	14.15	1.833	7-Hexadecene, (Z)-Molecular weight: 224.4253Other names: (7Z)-7-Hexadecene; cis-7-Hexadecene; (Z)-7-Hexadecene	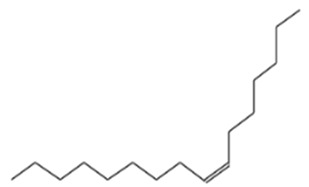	35507-09-6
Cyclopropane, nonyl-Molecular weight: 168.3190Other names: n-Nonyl-cyclopropane	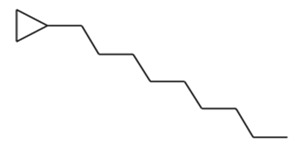	74663-85-7
3-Hexadecene, (Z)-Molecular weight: 224.4253Other names: (3Z)-3-Hexadecene; cis-3-Hexadecene; (Z)-3-Hexadecene	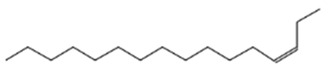	34303-81-6
5-Octadecene, (E)-Molecular weight: 252.4784Other names: (5E)-5-Octadecene; (E)-5-Octadecene; trans-5-Octadecene	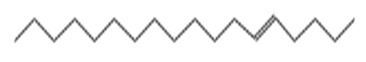	7206-21-5
9-Octadecene, (E)-Molecular weight: 252.4784Other names: (9E)-9-Octadecene; (E)-9-Octadecene; trans-9-Octadecene	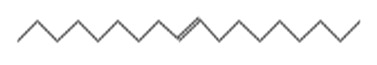	7206-25-9
5	14.76	1.330	DiphenylamineMolecular weight: 169.2224Other names: Benzenamine, N-phenyl-; Anilinobenzene; Benzene, (phenylamino)-; DFA; DPA; N-Phenylaniline; N-Phenylbenzeneamine; Aniline, N-phenyl-; Benzene, anilino-; Big Dipper; C.I. 10355; N-Phenylbenzenamine; N,N-Diphenylamine; No-Scald; Phenylaniline; Scaldip; Deccoscald 282; Difenylamin; N-Fenylanilin; No-Scald dpa 283; Naugalube 428 L; NSC 215210	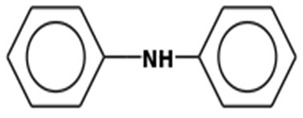	122-39-4
			2-p-TolylpyridineMolecular weight: 169.2224Other names: Pyridine, 2-(4-methylphenyl)-; 2-(4-Methylphenyl)pyridine	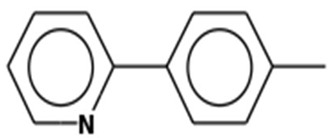	4467-06-5
			Hydrazine, tetraphenyl-Molecular weight: 336.4290Other names: Tetraphenylhydrazine	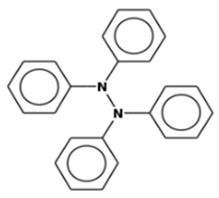	632-52-0
			Pyridine, 3-(phenylmethyl)-Molecular weight: 169.2224Other names: 3-Benzylpyridine; Pyridine, 3-benzyl-	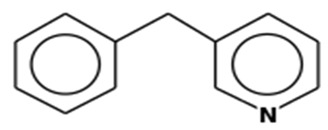	620-95-1
			1-Methyl-3,3-diphenylureaMolecular weight: 226.2738Other names: Urea, N′-methyl-N,N-diphenyl-; N,N-Diphenyl-N′-methylurea; 3-methyl-1,1-diphenylurea	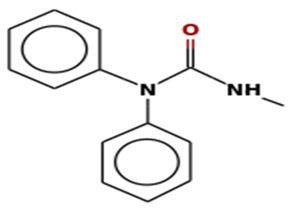	13114-72-2
6	16.35	3.822	n-Tridecan-1-olMolecular weight: 200.3608Other names: n-Tridecanol; n-Tridecyl alcohol; Tridecanol; 1-Hydroxytridecane; 1-Tridecanol; Tridecan-1-ol; Tridecyl alcohol	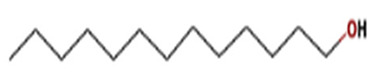	112-70-9
			5-Octadecene, (E)-Molecular weight: 252.4784Other names: (5E)-5-Octadecene; (E)-5-Octadecene; trans-5-Octadecene	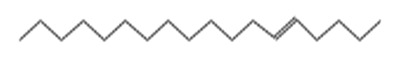	7206-21-5
			1-UndecanolMolecular weight: 172.3077Other names: Undecyl alcohol; n-Undecan-1-ol; n-Undecanol; n-Undecyl alcohol; Hendecanoic alcohol; Hendecyl alcohol; 1-Hendecanol; Alcohol c-11; n-Hendecylenic alcohol; Undecanol-(1); Tip-Nip; Undecanol; Decyl carbinol; Neodol 1; Undecan-1-ol; 1-Undecyl alcohol; NSC 403667	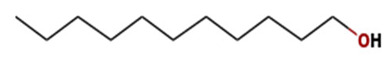	112-42-5
			9-Octadecene, (E)-Molecular weight: 252.4784Other names: (9E)-9-Octadecene; (E)-9-Octadecene; trans-9-Octadecene	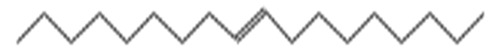	7206-25-9
			3-Octadecene, (E)-Molecular weight: 252.4784 Other names: (3E)-3-Octadecene; (E)-3-Octadecene; trans-3-Octadecene	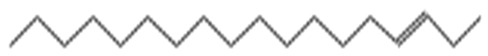	7206-19-1
7	18.35	2.036	3-Eicosene, (E)-Molecular weight: 280.5316Other names: (E)-icos-3-ene	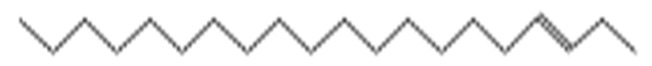	74685-33-9
			5-Octadecene, (E)-Molecular weight: 252.4784Other names: (5E)-5-Octadecene; (E)-5-Octadecene; trans-5-Octadecene	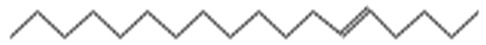	7206-21-5
			3-Octadecene, (E)-Molecular weight: 252.4784Other names: (3E)-3-Octadecene; (E)-3-Octadecene; trans-3-Octadecene	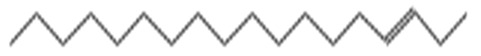	7206-19-1
			5-Eicosene, (E)-Molecular weight: 280.5316Other names: (5E)-5-Icosene; [E]-5-Eicosene	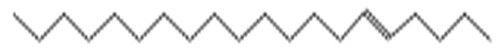	74685-30-6
			CeteneMolecular weight: 224.4253Other names: 1-Hexadecene; α-Hexadecene; n-Hexadec-1-ene; 1-Cetene; Hexadecylene-1; Hexadec-1-ene; Hexadecene-1; Neodene 16; 1-n-Hexadecene; NSC 60602	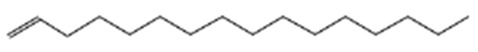	629-73-2
8	20.18	1.389	5-Eicosene, (E)-Molecular weight: 280.5316Other names: (5 E)-5-Icosene; [E]-5-Eicosene	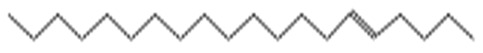	74685-30-6
			3-Eicosene, (E)-Molecular weight: 280.5316Other names: (E)-(E)-icos-3-ene; 3-Icosene; (3E)-3-Icosene	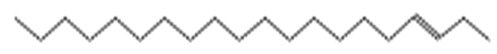	74685-33-9
			5-Octadecene, (E)-Molecular weight: 252.4784Other names: (5E)-5-Octadecene; (E)-5-Octadecene; trans-5-Octadecene	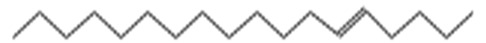	7206-21-5
			3-Octadecene, (E)-Molecular weight: 252.4784Other names: (3E)-3-Octadecene; (E)-3-Octadecene; trans-3-Octadecene	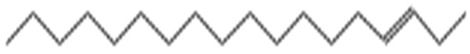	7206-19-1
			9-Eicosene, (E)-Molecular weight: 280.5Other names: (E)-(E)-icos-9-ene; 9-eicosene; icos-9-ene	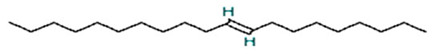	74685-29-3
9	21.87	0.814	Oxalic acid, allyl pentadecyl esterMolecular weight: 340.5Other names: 1-O-pentadecyl 2-O-prop-2-enyl oxalate	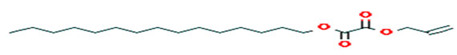	N/A
			5-Eicosene, (E)-Molecular weight: 280.5316Other names: (5E)-5-Icosene; [E]-5-Eicosene	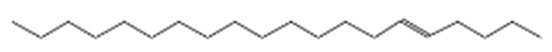	74685-30-6
			3-Eicosene, (E)-Molecular weight: 280.5316Other names: (E)-3-Icosene;3-Eicosene,(E)-; 3-Eicosene, (3E)-	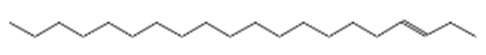	74685-33-9
			3-Octadecene, (E)-Molecular weight: 252.4784Other names: (3E)-3-Octadecene; (E)-3-Octadecene; trans-3-Octadecene	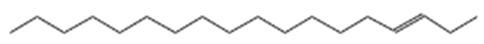	7206-19-1
			5-Octadecene, (E)-Molecular weight: 252.4784Other names: (5E)-5-Octadecene; (E)-5-Octadecene; trans-5-Octadecene	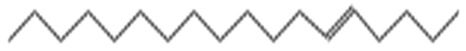	7206-21-5

## Data Availability

The data presented in this study are available on request from the corresponding author.

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
