# Peer review of "Pigmented Microbial Extract (PMB) from Exiguobacterium Species MB2 Strain (PMB1) and Bacillus subtilis Strain MB1 (PMB2) Inhibited Breast Cancer Cells Growth In Vivo and In Vitro"

_ijms, 2023, doi:10.3390/ijms242417412_

Round 1
Reviewer 1 Report
Comments and Suggestions for Authors
This is an interesting manuscript, and I would deem it to be of interest to the breast cancer community. In particular, Bandi et al. study the effects of pigmented microbial extracts (PMBs) on breast cancer cells' proliferation and migration. The authors assert that PMB1 effects are potentially mediated by the inhibition of Legumain (LGMN) activity and expression. Furthermore, the extract stimulated cancer cell apoptosis and blocked angiogenesis in EAC xenograft models. However, I feel that the manuscript, in sections, requires some considerable improvement to be deemed suitable for publication. The experiments are often not well justified and not well controlled with respect to what the experiments are trying to show. There needs to be a more robust comparison to draw any conclusion. So, more cell lines are clearly needed.
Below, I provide some suggestions for improving the manuscript:
1. In the text, there are paragraphs that clearly do not belong to the manuscript, for example, lines 49-57 and 399-402.
2. The images do not have good resolution and are difficult to understand.
3. Some images are overlapped, and the data are not legible, for example, Fig. 1.
4. In Fig. 1, which cell line was used for the experiments? In the Materials and Methods, RAW 264.7 is mentioned. Why? And what is the method used?
5. The initial experiments were performed in two negative breast cancer cell lines and one ERα+, PR+, Her2- (MCF7). In my opinion, more positive and triple-positive cell lines should be used to assert that PMB1 works better in MDA-MB-468. Furthermore, it would be appropriate to use MCF10A as a normal cell line.
6. In line 417, the authors claim that: Both extracts significantly inhibited LGMN activity in BC cells (Fig. 1). However, this statement is not correct because this data is not shown in Fig. 1.
7. In lines 418-419, the authors state: MDA-MB-468 cells demonstrated the highest LGMN expression and were chosen for further investigations. The authors should assess protein expression and gene levels of LGMN through Western blot and RT-PCR.
8. In the Materials and Methods, include the percentage of CO2 used for cell growth.
Comments on the Quality of English LanguageMinor editing of English language required
Author Response
"Please see the attachment."

Reviewer 2 Report
Comments and Suggestions for Authors
Author Response
"Please see the attachment."

Round 2
Reviewer 1 Report
Comments and Suggestions for Authors
This is an interesting manuscript, and I would deem it to be of interest to the breast cancer community. In particular, Bandi et al. study the effects of pigmented microbial extracts (PMBs) on breast cancer cells' proliferation and migration. The authors assert that PMB1 effects are potentially mediated by the inhibition of Legumain (LGMN) activity and expression. Furthermore, the extract stimulated cancer cell apoptosis and blocked angiogenesis in EAC xenograft models. The manuscript can be accept in the present form.